

# A learning-based synthesis approach of reward asynchronous probabilistic games against the linear temporal logic winning condition

Wei Zhao[1] and Zhiming Liu[2]

[1] College of Computer Science and Technology, Nanjing University of Aeronautics and Astronautics, Nanjing, Jiangsu, China
[2] School of Software, Northwestern Polytechnical University, Xi'an, Shaanxi, China

## ABSTRACT

The traditional synthesis problem is usually solved by constructing a system that fulfills given specifications. The system is constantly interacting with the environment and is opposed to the environment. The problem can be further regarded as solving a two-player game (the system and its environment). Meanwhile, stochastic games are often used to model reactive processes. With the development of the intelligent industry, these theories are extensively used in robot patrolling, intelligent logistics, and intelligent transportation. However, it is still challenging to find a practically feasible synthesis algorithm and generate the optimal system according to the existing research. Thus, it is desirable to design an incentive mechanism to motivate the system to fulfill given specifications. This work studies the learning-based approach for strategy synthesis of reward asynchronous probabilistic games against linear temporal logic (LTL) specifications in a probabilistic environment. An asynchronous reward mechanism is proposed to motivate players to gain maximized rewards by their positions and choose actions. Based on this mechanism, the techniques of the learning theory can be applied to transform the synthesis problem into the problem of computing the expected rewards. Then, it is proven that the reinforcement learning algorithm provides the optimal strategies that maximize the expected cumulative reward of the satisfaction of an LTL specification asymptotically. Finally, our techniques are implemented, and their effectiveness is illustrated by two case studies of robot patrolling and autonomous driving.

# INTRODUCTION

Reaction system synthesis is a technique that automatically explores how a system satisfies a specific specification (task) (*Bloem et al., 2012*). In recent years, with the rapid development of the intelligent industry, it has been widely used in robot patrolling, intelligent logistics, and intelligent transportation. However, it is still difficult to find a practically feasible synthesis algorithm and generate the optimal system. The construction

Corresponding author
Wei Zhao, wzhao618@nuaa.edu.cn

of a reactive system usually needs to produce outputs for inputs to fulfill the requirement that is typically described by a Linear Temporal Logic (LTL) formula. Constructing a correct reactive system needs to generate outputs for the inputs to fulfill some given specifications (*Buchi & Landweber, 1990*). This construction process can be graphically modeled as a two-player game between the system (outputs) and the environment (inputs) (*Rabin, 1972*). The goal of the game is to synthesize strategies for the player to satisfy a given LTL specification. The system win indicates that a given specification is satisfied. These games can be solved algorithmically, *i.e.*, one can determine which player wins the game and produce a winning strategy; the winner is guaranteed to have a strategy that is memoryless or only requires a finite memory.

Even though the synthesis of LTL specifications has been extensively studied, there are still some challenges. This is mainly because the system cannot respond in time or chooses the wrong behavior when interacting with the complex, changeable, and uncertain environment. In this case, with the further study of the system synthesis problem, the probability becomes the mainstream method to model and analyze reactive systems. Not only that, but probability theory has wider applications in the field of control (*Ren, Zhang & Zhang, 2019*; *Zhang & Wang, 2021*; *Liu, Zhang & Yue, 2021*). Recently, probabilistic synthesis is proposed and extensively studied, *e.g.*, (*Filar & Vrieze, 2012*; *Church, 1963*; *Shapley, 1953*; *Chatterjee, Henzinger & Jobstmann, 2008*; *Kwiatkowska & Parker, 2013*; *Neyman, Sorin & Sorin, 2003*; *Nilim & El Ghaoui, 2005*; *Lustig, Nain & Vardi, 2011*; *Dräger et al., 2014*). With the widespread attention to probabilistic synthesis, there is a growing interest in the problem of expected reward in the probabilistic environment in recent years. In a probabilistic environment, the value of a strategy for the system is the maximal reward of a play induced by this strategy, and the goal of the system is to maximize this value.

This study focuses on the synthesis problem of reactive systems and transforms it into the problem of probabilistic games with both LTL winning conditions and a reward mechanism. First, a learning-based approach is designed to motivate the system to satisfy the winning condition and generate strategies. In our work, uncertainty is considered both in the environment properties and in the system behaviors. Meanwhile, the reward properties are considered in our model. Based on this, this study establishes a probabilistic model with an asynchronous reward mechanism, which is referred to as reward asynchronous probabilistic game (RAPG). Then, based on the reinforcement learning method, algorithmic incentive systems are developed to win the game and generate corresponding strategies. To formulate synthesizing strategies for each player to maximize the expected cumulative rewards for satisfying the LTL objectives, this study first encodes reachability properties to obtain some sets of states that satisfy the specific requirement, then shrinks the state space, and constructs an asynchronous reward mechanism for players according to the winning condition. Finally, the asynchronous reward mechanism is combined with reinforcement learning to learn strategies that satisfy the given specifications.

To the best of our knowledge, there is no existing work on the strategy synthesis of RAPGs based on reinforcement learning. This study focuses on calculating the expected

cumulative reward for a play satisfying the winning condition under each state while maximizing the expected rewards against an adversarial probabilistic environment. The contributions of this article are summarized as follows:

- An asynchronous reward mechanism is designed to motivate the system to satisfy specific requirements, and a novel probabilistic model is proposed;
- Reachability properties are analyzed and encoded, which tremendously shrink the state space of the game;
- A learning-based approach is proposed to compute the maximum expected cumulative reward satisfying LTL specifications and generate corresponding strategies.

The rest of this article is organized as follows. In "Related work", an overview of the related work is given. The background definitions and notations are provided in "Preliminaries". "Synthesis problem through reward mechanism" introduces the definition of RAPGs, the synchronous reward mechanism, and the formalization of the research problem. The learning-based synthesis algorithm is introduced in "Synthesis algorithm based on reinforcement learning". The applicability of our algorithm is verified by using two examples in "Case study". Finally, "Summary and future work" concludes this article and discusses future research work.

## RELATED WORK

Reactive system synthesis under an uncertain environment is widely used in computer science, engineering, and economics, *e.g.*, autonomous driving and robot rescue operations (*Sutton & Barto, 2018*). Reactive systems have inherent complexity due to continuous interactions with the external environment. Meanwhile, the uncertainty of the environment brings a great challenge to system synthesis. The existing works on reactive synthesis problems mainly focus on environmental uncertainty, and the solution is to reduce the computational complexity of the synthesis algorithm (or optimize the synthesis algorithm) (*Bloem et al., 2012*; *Buchi & Landweber, 1990*; *Harding, Ryan & Schobbens, 2005*) and to repair unrealized specifications (*Hunt & Johnson, 2000*; *Könighofer, Hofferek & Bloem, 2013*; *Kuvent, Maoz & Ringert, 2017*; *Maoz, Ringert & Shalom, 2019*). In general, a requirement is defined as a contract about the assumption (input) on the environment behaviors and the guarantee (output) of the behavior of the system in a reactive system. That is, given the assumption (input) of the behavior of the environment, the system behavior is always guaranteed (output) to satisfy the specified specifications (*Zhao et al., 2022*). Unrealizable specifications can be understood as if the environment satisfies all assumptions while forcing the system behavior to violate some guarantees. The efficient synthesis algorithm and the method of repairing the unrealizable specifications do not directly analyze the influence of uncertainty on system synthesis. Different from these two methods, our work focuses on motivating systems to satisfy a given requirement by using the asynchronous reward mechanism.

As for probabilistic synthesis, most systems are modeled as reachability games, includes $2\frac{1}{2}$-player games (*Nilim & El Ghaoui, 2005*; *Svorenová & Kwiatkowska, 2016*), concurrent

games (*Neyman, Sorin & Sorin, 2003*; *De, Henzinger & Kupferman, 2007*), *etc*. *Nilim & El Ghaoui (2005)* proposed to model the complex system as a $2\frac{1}{2}$-player game. In general, the state space of a $2\frac{1}{2}$-player game involves a class of player states and a set of probabilistic states. *Kwiatkowska, Norman & Parker (2019)* introduces turn-based probabilistic timed multi-player games. Current games are also often used to model systems (*Hasanbeig et al., 2019*; *Kwiatkowska et al., 2021*). The characteristic of this game is that a state transition is performed through two actions taken by the player separately but independently. The games considered in this article differ from both $2\frac{1}{2}$-player games and concurrent games. In addition, *Almagor, Kupferman & Velner (2016)* proposed mean-payoff Markov Decision Processes (MDPs) with a parity winning condition to find a strategy that minimizes the expected cost of a play against a probabilistic environment. Our work considers modeling a reactive system as a RAPG. In each round of the RAPG, state transitions are determined alternately by two players who choose their actions and transitions. Meanwhile, both players will obtain corresponding rewards by choosing actions and transitions. Probabilistic reachability is an important property of APGs, which help to deeply study the calculation of the winning probability of the system. Another important property of PAPG that will be concerned in our work is the expected reachability. It allows using rewards and costs to model, *e.g.*, rewards for robots completing tasks, and safety drive for autonomous driving cars. Considering the reward property, this work is interested in calculating the cumulative rewards, *i.e.*, the sum of the rewards obtained when the system runs. Based on the calculation of the reward achieved on all runs of the system, the expected cumulative rewards can be obtained. Thus, the value of a strategy for the system is the maximal cumulative reward of a play induced by this strategy, and the goal of the game is to maximize this value by motivating the system to win.

Reinforcement learning (RL) designs algorithms to learn the optimal strategy which maximizes/minimizes the expected reward through interactions with the complex environment (*Sutton & Barto, 2018*). Typically, MDPs play a critical role in RL because of their unique ability to describe the time-independent state transition property. Generally, there are two types of RL algorithms: model-free algorithms and model-based algorithms (*Filar & Vrieze, 2012*; *Hasanbeig et al., 2019*; *Lavaei et al., 2020*; *Huh & Yang, 2020*; *Fu & Topcu, 2014*; *Brázdil et al., 2014*; *Puterman, 2014*). *Hasanbeig et al. (2019)* presents an approach to design optimal control strategies for Markov decision processes with unknown behavior by the model-free RL algorithm. This approach generates traces that satisfy specific LTL specifications with the maximized probability and returns the maximum expected reward. Many RL algorithms like *Q*-learning can be regarded as model-free algorithms. By updating the values of each state-action pair, these algorithms can directly learn the action-value function. Furthermore, the safe operation problem of the system can be solved by a model-free safety specification algorithm (*Huh & Yang, 2020*). The method is to learn the maximum probability of safe operations by combining probabilistic reachability with a safe RL algorithm. Model-based reinforcement learning algorithms are also often used to design strategies, such as in *Fu & Topcu (2014)*, *Brázdil et al. (2014)*, to synthesize strategies that maximize the satisfaction probability

for Markov decision processes (MDPs). Most of these studies model the environment as an MDP and an extended MDP. *Brázdil et al. (2014)* proposed the expected total reward for discrete-time Markov chains (DTMCs) by solving a set of equation systems and for MDPs by solving a linear program (*Filar & Vrieze, 2012*). In our work, formal methods are combined with learning-based methods to explore the reward properties of probabilistic synthesis. In particular, for APGs, this work focuses on RL algorithms to compute the expected cumulative reward of the system winning. Precisely, the RL method is used to learn the strategy to motivate the system to win and compute the expected cumulative reward for each player.

## PRELIMINARIES

This section briefly introduces the definitions used in our article, including the specification language, games, and a modal $\mu$-calculus over asynchronous probabilistic games. LTL is taken as the desired specification language, and LTL specifications are first introduced.

### LTL

LTL has been increasingly popular as a tool to describe specific requirements when synthesizing strategies for reactive systems.

**Syntax.** Given a finite and non-empty set $\mathscr{V}$ of atomic propositions, the arbitrary proposition in $\mathscr{V}$ is denoted as $p$. Given a position, Boolean variables have a unique truth value as True or False. Note that temporal operators usually have two conventional notations, either $\mathsf{X}, \mathsf{G}, \mathsf{F}, \mathsf{U}$ or $\bigcirc, \square, \diamond, \mathscr{U}$. In this article, the former is followed.

An LTL formula $\psi$ is defined inductively according to the following grammar:

$$\pi ::= True | \neg\psi | \psi_1 \vee \psi_2 | \mathrm{X}\psi | \psi_1 \mathrm{U}\psi_2$$

where the Boolean constants *True* and *False* can be denoted by formulas "$\top$" and "$\perp$", respectively; $\neg$ and $\vee$ are the logic connectives *negation* and *disjunction*; $\mathsf{X}$ and $\mathsf{U}$ are the temporal operators *next* and *until*. If $\psi$ is a LTL formula, $\neg\psi$ is also a LTL formula. In addition, logic connectives *conjunction* ($\wedge$), *implication* ($\Rightarrow$), and *equivalence* ($\Leftrightarrow$) can be defined as $\psi_1 \wedge \psi_2 \equiv \neg(\neg\psi_1 \vee \neg\psi_2), \psi_1 \Rightarrow \psi_2 \equiv \neg\psi_1 \vee \psi_2$, and $\psi_1 \Leftrightarrow \psi_2 \equiv (\psi_1 \Rightarrow \psi_2) \wedge (\psi_2 \Leftrightarrow \psi_1)$, respectively. Additional temporal operators such as *eventually* ($\mathsf{F}$) and *always* ($\mathsf{G}$) are derived as $\mathsf{F}\psi \equiv \top\mathsf{U}\psi$ and $\mathsf{G}\psi \equiv \neg\mathsf{F}\neg\psi$.

**Semantics.** Given a finite set of Boolean variables $\mathscr{V}$, an infinite sequence $\pi = \pi_0\pi_1 \cdots \in (2^{\mathscr{V}})^\omega$ is defined as a computation. Denote the LTL formula that $\psi$ holds at position $i \geq 0$ of $\pi$ as $\pi, i \models \psi$, which is the satisfaction relation. The semantics of LTL formulas are formally defined as follows:

- $\pi, i \models p$ if and only if $p \in \pi_i$
- $\pi, i \models \neg\psi$ if and only if $\pi, i \nvDash \psi$
- $\pi, i \models \psi_1 \vee \psi_2$ if and only if $\pi, i \models \psi_1$ or $\sigma, i \models \psi_2$
- $\pi, i \models \mathrm{X}\,\psi$ if and only if $\pi, i+1 \models \psi$

- $\pi, i \models \psi_1 U \psi_2$ if and only if there exists $k \geq i$ such that $\pi, k \models \psi_2$ and for all $i \leq j < k$, $\pi, j \models \psi_1$

Intuitively, $X\psi$ means that $\psi$ holds (or is true) in the next position (or the next "step") in the computation; $\psi_1 U \psi_2$ means that $\psi_1$ holds until $\psi_2$ becomes True. The computation $\pi$ satisfies $\psi$ if $\pi, 0 \models \psi$, which is denoted as $\pi \models \psi$. If $\psi$ satisfies $\pi$ in every position of the computation, it means that $\pi$ satisfies $A\psi$; if $\psi$ will be satisfied at least once in the future, then the computation $\pi$ satisfies $F\psi$. Besides, this article defines $\pi, i \models GF\psi$ if $\psi$ will be true infinitely times in the computation, and $\pi, i \models FG\psi$ if $\psi$ will eventually be continuously true start from some position in the computation. Meanwhile, $GF\psi$ is usually used to denote the goal of systems or environments that need to be satisfied.

## Asynchronous probabilistic games

The interaction between the system and the environment is transformed into an asynchronous probabilistic game, which helps to analyze the uncertainty when the system interacts with the environment. Now, the definitions of the asynchronous probabilistic games are introduced below.

**Definition 1.** *An Asynchronous Probabilistic Game (APG) is defined as a tuple $\mathcal{G} = \langle \mathcal{V}, A_{ct}, V, P_e, P_s, L \rangle$, where $\mathcal{V}$ is a finite set of atomic propositions, and $\mathcal{V}$ is the set of states on the game arena. Let $V = V_e \cup V_s$ and $V_e \cap V_s = \varnothing$, where $V_e$ and $V_s$ are the sets of environment states and system states, respectively. $P_e : V_e \times A_{ct} \to Dist(V_s)$ is a transition function of the environment such that $P(v_e, a)(v_s)$ is the probability to transit from environment state $v_e$ to system state $v_s$ on taking action $a$, where $Dist(V_s)$ is a discrete probability distribution over $V_s$. $P_s : V_s \times A_{ct} \to Dist(V_e)$ is a transition function of the system such that $P(v_s, a)(v_e)$ is the probability to transit from system state $v_s$ to environment state $v_s$ on taking action $a$, where $Dist(V_e)$ is a discrete probability distribution over $V_e$. $L : V \to 2^{\mathcal{V}}$ is a labeling function, and $L(v)$ is a set of atomic propositions that holds in $v$, where $v \in V$.*

In a game, the steps are executed alternatively by the environment and the system. Given an APG $\mathcal{G}$, a finite (or an infinite) sequence $\pi = v_0, v_1, \cdots$ is the play (or path) of $\mathcal{G}$ if for each $i >= 0$, $v_{i+1}$ is a successor of $v_i$. That is, if $v_i \in V_e$, $P_e(v_i, a)(v_{i+1}) > 0$ for some $a \in A_{ct}$ and $P_s(v_i, b)(v_{i+1}) = 0$ for all $b \in A_{ct}$, and if $v_i \in V_s$, $P_e(v_i, a)(v_{i+1}) = 0$ for all $a \in A_{ct}$ and $P_s(v_i, b)(v_{i+1}) > 0$ for some $b \in A_{ct}$. Let $v_0$ be the initial state of $\pi$ and $\Pi$ be the set of all plays of $\mathcal{G}$. For a state $v \in V$, $\Pi^v$ is the set of plays with $v$ as the initial state.

Given a game $\mathcal{G}$ and a LTL formula $\psi$, this work uses $\psi$ to denote the winning condition of the game $\mathcal{G}$. Let $\pi = v_0, v_1, \cdots$ be a play, $\pi$ is winning for the system under a given winning condition $\psi$ if $\pi$ is a finite play and the last state $v_n$ is the environment state in which there is no action $a \in A_{ct}$ such that $P_e(v_n, a)(v_s) > 0$, or $\pi$ is an infinite play and $\pi$ satisfies the winning condition $\psi$; otherwise, it is said that $\pi$ is winning for the environment.

For an APG $\mathcal{G}$ and a state $c$ of the set $V$, an action $a \in A_{ct}$ is in state $c$ if there is a state $d \in V$ such that $P_e(c, a)(d) = 1$. This article denotes the set of actions in $c$ as $A_{ct}(c)$.

Assume that $Next_s$ and $Next_e$ are the sets of finite plays with the last state in $V_s$ and $V_e$, respectively.

For an APG $\mathcal{G}$, a strategy for the system of $\mathcal{G}$ is a function $f : Next_s \rightarrow A_{ct}$, and $f(v_0 \ldots v_n) \in Act(v_n)$ is the next action to choose by the system. Similarly, a strategy can be defined for the environment. This article denotes the sets of the system strategies and environment strategies of $\mathcal{G}$ as $F_s$ and $F_e$, respectively. A strategy is memoryless if it relies only on the current state of the play and is not related to the history of the play. Formally, for any $\pi_1$ and $\pi_2$ in $Next_s$ (or $Next_e$) and any state in $V_s$ (or $V_e$), we have $f(\pi_1 v) = f(\pi_2 v)$. In addition to memoryless strategies, there are also history-dependent strategies and finite-memory strategies. In this article, it is sufficient to consider only memoryless strategies (*Kwiatkowska & Parker, 2013*).

Given a play $\pi = v_0 v_1 \ldots v_i \ldots$ that follows a system (or environment) strategy $f$, let each finite prefix $\tau = v_0 v_1 \ldots v_i \in Next_s$ (or $Next_e$), and we have $P(\tau, f(\tau))(v_{i+1}) > 0$. Let $\psi$ be a state proposition and $v$ be a state in $V$, this article denotes the probability that the play's initial states satisfies $\psi$ and follow strategy $f$ as $Pr_f(v \models \psi)$.

For a game $\mathcal{G}$, let $T \subseteq V$ be a set of states that satisfy Boolean expression $\psi$ (or a state proposition in general), *i.e.*, there is a state $v \in T$ such that $\psi$ is true of $v$. This article uses $T$ to denote a set of states that satisfy $\psi$, if $\psi$ is the winning condition. $T$ is the set of target states in game $\mathcal{G}$, where any play starting from any state in $T$ satisfies the winning condition $\psi$.

Next, this article defines the reachability probability property over the game $\mathcal{G}$. Before the definition is given, the reachability and fairness properties of $T$ are explained. Given an LTL formula F$\psi$, it means that $\psi$ holds in some state of the computation. The reachability property of $T$ is that some states in $B$ occur in the computation of the game. An LTL formula GF$\psi$ means that $\psi$ holds for infinite time in the computation. Then, the fairness property of $T$ is that some states in set $T$ occur infinite times in the computation.

**Definition 2.** *Given an APG $\mathcal{G}$, $f$ is a strategy, $\psi$ is a winning condition, and $T \subseteq V$ is a set of states that satisfies $\psi$. For a state $v \in V$, this article uses $v \models \text{F}T$ to denote a play that starts from $v$, satisfies $\psi$, and reaches some states in $T$. Meanwhile, this article uses $Pr(v \models \text{F}T)$ to denote the probability of this type of play. If the play also follows $f$, the probability of the play follow $f$ is denoted as $Pr_f(v \models \text{F}T)$.*

## A variant of modal $\mu$-calculus over APGs

Most modal/temporal logic can be viewed as a sub-logic of $\mu$-Calculus, where a powerful extension is modal $\mu$-Calculus. This logic is succinct in syntax, and formula variables are often used in such logic. The semantics of a $\mu$-calculus formula is defined by the Kripke structure, which designates the set of states that satisfy the formula (*Kesten, Piterman & Pnueli, 2005*). This article defines the variant of modal $\mu$-calculus over the APGs structure:

Given an APG structure $\mathcal{G} : \langle \mathcal{V}, A_{ct}, V, P_e, P_s, L \rangle$. For every state $v \in V$, the formulas $p$ and $\neg p$ are atomic formulas of $\mathcal{G}$. Let $Var = \{M, N, \ldots\}$ be a set of formula variables. The syntax of $\mu$-calculus formulas is defined by the following grammar:

$$\psi ::= p|\neg p|X|\psi_1 \vee \psi_2|\psi_1 \wedge \psi_2|\circledast\psi|\odot\psi|\mu X\psi|\nu X\psi$$

A formula $\psi$ is described as the set of $\mathcal{G}$-states in which $\psi$ is true. This article uses $[[\psi]]_{\mathcal{G}}^{\varepsilon}$ to denote such a set of states, indicating that the set satisfies $\psi$ under $\varepsilon$. Here, $\varepsilon : Var \rightarrow 2^{\mathcal{V}}$ is an assignment that assigns formula variables to sets of atomic propositions in $\mathcal{V}$. The set $[[\psi]]_{\mathcal{G}}(\varepsilon)$ is inductively defined as follows:

- $[[p]]_{\mathcal{G}}(\varepsilon) = \{v \in V|p \in L(v)\}$.
- $[[\neg p]]_{\mathcal{G}}(\varepsilon) = V\backslash[[p]]_{\mathcal{G}}(\varepsilon)$.
- $[[X]]_{\mathcal{G}}(\varepsilon) = \varepsilon(X)$.
- $[[\psi_1 \vee \psi_2]]_{\mathcal{G}}(\varepsilon) = [[\psi_1]]_{\mathcal{G}}(\varepsilon) \cup [[\psi_2]]_{\mathcal{G}}(\varepsilon)$.
- $[[\psi_1 \wedge \psi_2]]_{\mathcal{G}}(\varepsilon) = [[\psi_1]]_{\mathcal{G}}(\varepsilon) \cap [[\psi_2]]_{\mathcal{G}}(\varepsilon)$.
- 
$$[[\circledast\psi]]_{\mathcal{G}}(\varepsilon) = \left\{ \begin{array}{c} m \in V_e|\forall a \in A_{ct}(m), for\ all\ n \in V_s : \\ P_e(m,a)(n) > 0 \Rightarrow n \in [[\psi]]_{\mathcal{G}}(\varepsilon) \end{array} \right\}$$
$$\cup \left\{ \begin{array}{c} m \in V_s|\exists a \in A_{ct}(m), for\ all\ n \in V_e : \\ P_s(m,a)(n) > 0 \Rightarrow u \in [[\psi]]_{\mathcal{G}}(\varepsilon) \end{array} \right\}$$

A state $m$ is included in $[[\circledast\psi]]_{\mathcal{G}}(\varepsilon)$. If $m$ is the environment state, it can choose any action to reach a state in $[[\psi]]_{\mathcal{G}}(\varepsilon)$; if $m$ is the system state, it can choose an appropriate action to move into $[[\psi]]_{\mathcal{G}}(\varepsilon)$.

- 
$$[[\odot\psi]]_{\mathcal{G}}(\varepsilon) = \left\{ \begin{array}{c} m \in V_e|\exists a \in A_{ct}(m), for\ all\ n \in V_s : \\ P_e(m,a)(n) > 0 \Rightarrow n \in [[\psi]]_{\mathcal{G}}(\varepsilon) \end{array} \right\}$$
$$\cup \left\{ \begin{array}{c} m \in V_s|\forall a \in A_{ct}(m), for\ all\ n \in V_e : \\ P_s(m,a)(n) > 0 \Rightarrow u \in [[\psi]]_{\mathcal{G}}(\varepsilon) \end{array} \right\}$$

A state $m$ is included in $[[\odot\psi]]_{\mathcal{G}}^{\varepsilon}$. If $m$ is an environment state, it chooses an appropriate action to move into $[[\psi]]_{\mathcal{G}}(\varepsilon)$; if $m$ is a system state, it can choose any action to reach a state in $[[\psi]]_{\mathcal{G}}(\varepsilon)$.

- $[[\mu X\psi]]_{\mathcal{G}}(\varepsilon) = \bigcup_i S_i$ where $S_0 = \varnothing$ and $S_{i+1} = [[\varphi]]_{\mathcal{G}}(\varepsilon[X/S_i])$.
- $[[\nu X\psi]]_{\mathcal{G}}^{\varepsilon} = \bigcap_i S_i$ where $S_0 = V$ and $S_{i+1} = [[\psi]]_{\mathcal{G}}(\varepsilon[X/S_i])$.

In addition, based on the syntax of the modal $\mu$-calculus formula, the following formulas can be further derived:

- $F_{\circledast}T = \mu Z.(\circledast Z \cup T)$ is a set with the characteristic that if state $v \in V$ is a system state, then all successors of $v$ are in $F_{\circledast}T$; if state $v$ is an environment state, then there exists an action such that a successor of $v$ in $F_{\circledast}T$. Especially, if $v$ is in $T$, it must be in $F_{\circledast}T$, whether $v$ is an environment state or a system state.
- $G_{\odot}T = \nu B.(\odot B \cap T)$is a set. If state $v \in V$ is an environment state and it is in set $T$, then all successors of $v$ are in $G_{\odot}T$; if state $v$ is a system state, there exists an action such that a successor state of $v$ in $G_{\odot}T$.

- AG EF$T = vB.[\mu.Y(\oplus Y \cup T) \cap \otimes B]$ is a set. If $v \in$ AG EF$T$, any path from $v$ can reach $T$ infinitely times, where $\otimes$ is the AX operator of CTL, $\oplus$ is the EX operator of CTL.

Based on the above $\mu$-calculus formulas, the analysis of the state space of the game is presented in "Synthesis algorithm based on reinforcement learning".

# SYNTHESIS PROBLEM THROUGH REWARD MECHANISM

In this section, the synthesis problem is defined based on reinforcement learning that stochastically approximates the value function of a probabilistic game. This article mainly focuses on the expected cumulative rewards for the system winning. The interaction of the system is first modeled with its environment as a reward asynchronous probabilistic game (RAPG), and it is defined as follows.

First, given an APG

$$\mathscr{G} = \langle \mathscr{V}, A_{ct}, V, P_e, P_s, L \rangle$$

and a linear temporal logic specification $\psi$.

This article uses rewards as additional quantitative measures of the APG to stimulate the system to win the game. Although some researchers use cost mechanisms to describe minimization (*e.g.*, energy consumption), reward mechanisms are commonly used to suggest a property that describes maximization (*e.g.*, profit) (*Nilim & El Ghaoui, 2005*). In our work, the reward mechanism involves attaching a reward value to the positions and actions available in each state to motivate the player to win, and the reward accumulates over transitions. Formally, the rewards mechanism of an APG is defined as follows:

**Definition 3.** *The reward mechanism for an APG $\mathscr{G} = \langle \mathscr{V}, A_{ct}, V, P_e, P_s, L \rangle$ is described as a specifying asynchronous reward structure, which is a tuple $R = (R_{st}, R_{ac})$ composed of two reward functions. One is state reward function $R_{st} : V \rightarrow R$ that maps the state $v$ of $\mathscr{G}$ to non-negative reals, and other is an action reward function $R_{ac} : V \times A_{ct} \rightarrow R$ that maps state-action pairs $(v, a)$ of $\mathscr{G}$ to non-negative reals, where $v \in V, a \in A_{ct}$.*

The action reward in a synchronous reward mechanism is also called a transition reward, impulse reward, or state-action reward. By definition, reward functions are Markovian, and they typically map states, or states and actions, to a scalar reward value.

Based on an APG and the asynchronous reward mechanism over it, thus article transforms the interaction of the system with its environment as a reward asynchronous probabilistic game (RAPG), which is a seven-tuple as follows:

$$\mathscr{G} = \langle \mathscr{V}, A_{ct}, V, P_e, P_s, L, R \rangle.$$

In *Zhao et al. (2022)*, the probabilistic reachability property is defined over APGs. This article discusses the expected reward properties that are defined over the above model RAPG. The model RAPG is an extension of APG and has all the properties of APG. Define the cumulative reward property over RAPG as:

**Definition 4.** *Let $\mathscr{G}$ be a RAPG and V be the state space over RAPG. For the winning condition $\psi$ and a set $T \subseteq V$, all plays starting from any states in T satisfy $\psi$. For an infinite*

*path $\pi = v_0, v_1, \cdots$ of the game $\mathcal{G}$, the cumulative reward for synchronous reward structure $R$ along an infinite path $\pi = v_0, v_1, v_2, \cdots$ is*

$$R(\pi) = \sum_{n=0}^{\infty} [R_{st}(v_n) + R_{ac}(v_n, a_n)].$$

*For an finite path $\pi$, define*

$$R(\pi, \text{FT}) = \sum_{i=0}^{i=n-1} [R_{st}(v_i) + R_{ac}(v_i, a_i)],$$

*if $v_i \notin T$ for $0 \le i \le n$ and $v_n \in T$;*

$$R(\pi, \text{FT}) = 0$$

*if $\pi \nvDash \text{FT}$.*

Note that $R(\pi, \text{FT})$ denotes the cumulative reward earned along an infinite path $\pi$ until some states in $T$ are reached for the first time. Cumulative rewards property is usually used to handle the sum of rewards accumulated from a position (or state) to a specific position (or state). Meanwhile, many other reward-based properties can be defined over RAPGs, such as discounted reward and expected long-run average reward. The characteristic of discount rewards is that the reward gain in each step is the reward multiplied by a discount factor $\lambda$ (in general $\lambda < 1$), so the strategy with fewer steps is generally preferred. Different from the discount reward, the expected long-run average reward considers the average reward earned in each state or transition.

**Definition 5.** *Given a RAPG $\mathcal{G}$, $V$ is the set of state spaces over the game $\mathcal{G}$; $f$ is a strategy, $\psi$ is a winning condition; $T \subseteq V$ is a specific set, and any play $\pi$ starting from a state in $T$ satisfies $\psi$; $R = (R_{st}, R_{ac})$ is a synchronous reward structure. This article denotes the expected cumulative reward of a play that starts from $v \in V$ and satisfies $\psi$ until reaching some states in $T$ as*

$$ER(v \models \text{FT}).$$

If the play follows $f$, this article uses $ER_f(v \models \text{FT})$ to represent its expected cumulative reward.

In addition, if $Pr(v \models \text{FT}) = 0$, then

$$ER_f(v \models \text{FT}) = 0;$$

otherwise, if $Pr(v \models \text{FT}) \ne 0$, then:

if $v \in V_e$, this article uses $ER^{min}(v \models \text{FT}) = \inf_{f \in F} ER_f(v \models \text{FT})$ to denote the minimal expected cumulative reward;

if $v \in V_s$, this article uses $ER^{max}(v \models \text{FT}) = \inf_{f \in F} ER_f(v \models \text{FT})$ to denote the maximal

expected cumulative reward. $F$ is a set of strategies for the system and environment over RAPG $\mathcal{G}$.

| **Algorithm 1: Computing the set of states $v$ with $ER(v \models FB) = 0$** |
|---|
| 1: Input: A RAPG $\mathcal{G}$ with finite space $V$ and a state set $T \subseteq V$ |
| 2: $B = V$ |
| 3: while $\odot B \cap V\backslash T \neq B$ do |
| 4: $B = \odot B \cap V\backslash T$ |
| 5: endwhile |
| 6: output: $G_\odot \neg T : \{v|v \in V \wedge ER(v \models FT) = 0\}$ |

This article is interested in computing either the maximum or (and) the minimum value of the cumulative expected reward of a play. This problem is formally defined as follows:

**Definition 6.** *For a RAPG $\mathcal{G}$, $V$ is the set of state spaces over the game $\mathcal{G}$; $f$ is a strategy; $\psi$ is a winning condition; $T \subseteq V$ is a specific set, and any play $\pi$ starting from a state in $T$ satisfies $\psi$; $R = (R_{st}, R_{ac})$ a synchronous reward structure. To stimulate the system to win, a strategy $f$ that maximizes (or minimizes) the expected cumulative reward $ER(v \models FB)$ of the system (or environment) and satisfies $\psi$ should be synthesized.*

# SYNTHESIS ALGORITHM BASED ON REINFORCEMENT LEARNING

This section discusses the problem of finding the learning-based synthesis algorithm based on the RAPG $\mathcal{G}$. To compute the expected cumulative reward, this article first analyzes and discusses how to divide the state space of the RAPG $\mathcal{G}$. Then, a highly efficient incremental probabilistic synthesis algorithm based on reinforcement learning is proposed.

## Qualitative reachability

Consider a RAPG $\mathcal{G}$, the winning condition $\psi$, and a set of states $T \subseteq V$, where all plays starting from any states in $T$ satisfies $\psi$. This article first uses $\mu$-calculus formulas to obtain a state set $G_\odot \neg T$, in which all plays starting from any states of the do not satisfy $\psi$. Specifically, for $v \in G_\odot \neg T$, (a) there is at least one successor of $v$ not in $T$ if $v$ is an environment state; (b) the successors of $v$ are all not in $T$ if $v$ is a system state. If $v \in G_\odot \neg T$, and then $Pr(v \models FT) = 0$. Especially, the expected cumulative reward of all plays starting from the state in set $G_\odot \neg T$ is 0. That is, if $v \in G_\odot \neg T$, then $ER(v \models FT) = 0$. The set $G_\odot \neg T$ can be computed by using Algorithm 1. By analyzing the state space, computing abstractions of the whole state space can be effectively avoided.

To compute the expected cumulative reward, this article does not need to consider the sure-reachability play starting from the state $v$ ($v \in V$) because it also needs some rewards to reach $T$, unless $v \in T$.

## Synthesis through reinforcement learning

The goal of synthesis is to incentivize the system to satisfy the LTL winning condition through a reward mechanism. This goal can be achieved by computing the maximum expected cumulative reward for each state. Meanwhile, some standard techniques in the

reinforcement learning literature can be adopted to find the satisfying strategies. Below, the calculation method is introduced in detail:

**Theorem 1.** *Consider an RAPG $\mathscr{G}$, the state space $V = V_E \cup V_S$ over $\mathscr{G}$, the winning condition $\psi$, and a set of states $T$ that satisfies $\psi$, i.e., all plays that start with the state in $T$ as the initial state satisfies $\psi$, and an asynchronous reward structure $R = (R_{st}, R_{ac})$, where $T \subseteq V$. Let $x_v^k$ denote the maximizing value of the expected cumulative reward under strategy $f$ in state $v$, where $k \geq 0$ is the expected cumulative reward parameter. For the convenience and simplicity of expressing $ER_f(v \models \text{F}T)$, the definition $x_v^k$:*

$$x_v^k := ER_f(v \models \text{F}T)$$

*is given. To compute the expected cumulative reward, there is no need to consider the state in $B$, and define $x_v^0 = 0$ for all $v \in V$. This simplifies the definition of the value $x_v^k$ for each state:*
  *if $v \in V_s$, then*

$$x_v^k = R_{st}(v) + \max_{a \in A_{ct}}(R_{ac}(v, a) + \sum_{v' \in V_e} P_S(v, a)(v') \cdot x_{v'}^{k-1});$$

  *if $v \in V_e$, then*

$$x_v^k = R_{st}(v) + \min_{a \in A_{ct}}(R_{ac}(v, a) + \sum_{v' \in V_s} P_e(v, a)(v') \cdot x_{v'}^{k-1}).$$

Algorithm 2 below is a learning-based algorithm that computes the expected cumulative reward for each state and extracts strategies. The algorithm has good scalability.

## CASE STUDY

In this section, two case studies are presented to illustrate our synthesis method. One is the problem of robot patrolling in a certain area, and the other is the problem of safety reachability of unmanned cars.

### Robot patrolling

As shown in Fig. 1, the robot performs the task of patrol in an area, and this area is divided into four regions. The robot patrol route starts at region 1 and goes through region 2 and then region 3 to region 4. In this scenario, if it encounters a person in region 2 and region 3, the robot will stay in that region with the person. Meanwhile, if it encounters an unknown hazardous item, the robot will pick it up and take it to region 4. Furthermore, it is assumed that a hazardous item and people do not appear at the same time. If a hazardous item appears first, then people will not appear in the process of delivering a hazardous item. Even if a second hazard item appears, the robot sends the first hazard item to region 4 before processing the next task.

The game graph corresponding to Fig. 1 is illustrated in Fig. 2. Consider an *RAPG* $\mathscr{G} = \langle \mathscr{V}, A_{ct}, V, \Theta_e, \Theta_s, L, R \rangle$, where $V_e = \{v_0, v_2, v_4, v_6\}$ is the set of environment states, $V_s = \{v_1, v_3, v_5, v_7\}$ is the set of system states, $A_{ct} = \{a, b, c, d, e\}$ is the set of actions, and $R = (R_{st}, R_{ac})$ is the synchronous reward structure. At the environment state, the environment can choose to put items and appear people to affect robot patrol. If the item is

---

**Algorithm 2: Learning-based algorithm for RAPG $\mathscr{G}$**

1: input: An RAPG $\mathscr{G}$ with finite space $V$, a state set $T \subseteq V$, and $G_{\odot}\neg T$

2: $\forall v \in V : x_v^0 = 0$;

3: **for** $k = 1, 2, \cdots$

4: **if** $v \in T \cup G_{\odot}\neg T$

5: $x_v^k = 0$

6: **else**

7: $\forall v \in V \setminus T$:

8: **if** $v \in V_S$ **then**

9: $x_v^k = R_{st}(v) + \max_{a \in A_{ct}}(R_{ac}(v, a)) + \sum_{v' \in V_e} P_s(v, a)(v') \cdot x_{v'}^{k-1}$;

10: **endif**

11: **else**

12: **if** $v \in V_e$ **then**

13: $x_v^k = R_{st}(v) + \min_{a \in A_{ct}}(R_{ac}(v, a)) + \sum_{v' \in V_e} P_e(v, a)(v') \cdot x_{v'}^{k-1}$;

14: **end if**

15: **if** $\max_{v \in V}|x_v^k - x_v^{k-1}| < \theta$

16: **break**

17: **else**

18: $x_v^k = x_v^{k-1}$

19: **endif**

20: **endfor**

21: output: the expected cumulative reward $x_v^k$ for all $v \in V$, optimal strategies $f$.

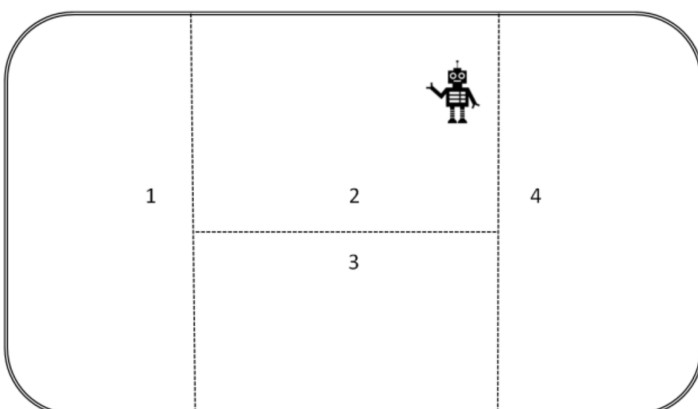

**Figure 1  Working area division of the robot.**

the hazard item, the robot will bring it to region 4. These two actions are denoted as $a$ and $b$, respectively. At this time, the system can choose to pick up or stay, and these two actions are denoted as $c$ and $d$, respectively. In this case, the robot identify the item as a hazard item and picks it up, which is considered the same action.

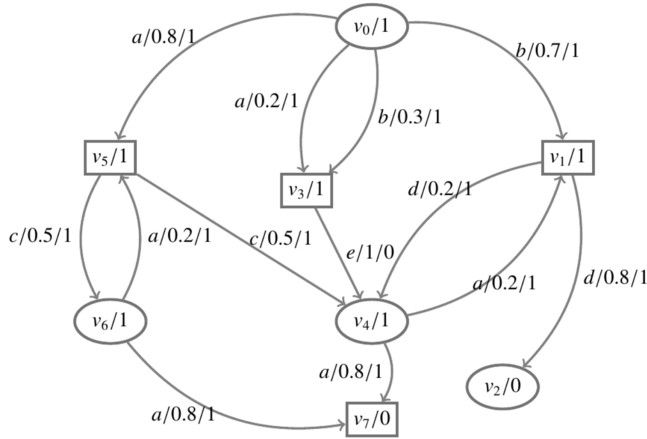

**Figure 2 The environment and the system are depicted as circles and squares, respectively.** The set
$T = \{v_7\}$. $AP = \{stay, arr\}$ is the set of atomic propositions.

The game proceeds as follows: Starting from the initial state $v_0$, the robot patrols normally and at region 1. When the environment chooses action $a$, the probability that a hazard item appears is 0.8, and the probability that a hazard item appear does not appear is 0.2. When the environment chooses action $b$, the probability that a person appears is 0.7, and the probability that no person appears is 0.3. When the robot (the system) selects action $c$, the probability of picking up the hazard item is 0.5, and the probability of not picking up the hazard item is 0.5. When the robot (the system) selects action $d$, the probability of staying in the region is 0.8, and the probability of leaving the region is 0.2. If the robot patrols normally, it is considered that the robot chooses action $e$, and the probability of normal patrol is 1. Figure 2 shows the specific game of the environment and the system, where $v_2$ is the stay state with the tag *stay*, $v_7$ is the state when the robot takes the hazard item to region 4, and $v_7$ is the target arrival state with the tag *arr*. So, for the winning condition $\psi$ and set $T = \{v_7\}$, all plays starting from a state in $T$ satisfies $\psi$, and $G_\odot \neg T = \{v_2\}$. Consider the RAPG from Fig. 2, the asynchronous reward structure $R$ assigns $R(v_0) = R(v_1) = R(v_3) = R(v_4) = R(v_5) = R(v_6) = 1$, $R(v_2) = R(v_7) = 0$, $R(v_0, a) = R(v_0, b) = R(v_1, d) = R(v_4, a) = R(v_5, c) = R(v_6, a) = 1$, and $R(v_3, e) = 0$.

According to Algorithm 2, the expected cumulative reward is computed to reach $T$. Below, the value of $x_v^k$ is presented for each state:

$k = 0: x_{v_0}^0 = 0, x_{v_1}^0 = 0, x_{v_2}^0 = 0, x_{v_3}^0 = 0, x_{v_4}^0 = 0, x_{v_5}^0 = 0, x_{v_6}^0 = 0, x_{v_7}^0 = 0$

$k = 1: x_{v_0}^1 = 2, x_{v_1}^1 = 2, x_{v_2}^1 = 0, x_{v_3}^1 = 1, x_{v_4}^1 = 2, x_{v_5}^1 = 2, x_{v_6}^1 = 2, x_{v_7}^1 = 0$

$k = 2: x_{v_0}^2 = 3.7, x_{v_1}^2 = 2.4, x_{v_2}^2 = 0, x_{v_3}^2 = 3, x_{v_4}^2 = 2.4, x_{v_5}^2 = 4, x_{v_6}^2 = 2.4, x_{v_7}^2 = 0$

. . .

$k = 8: x_{v_0}^8 = 4.80, x_{v_1}^8 = 2.50, x_{v_2}^8 = 0, x_{v_3}^8 = 3.50, x_{v_4}^8 = 2.50, x_{v_5}^8 = 4.93, x_{v_6}^8 = 2.98, x_{v_7}^8 = 0$

$k = 9: x_{v_0}^9 = 4.80, x_{v_1}^9 = 2.50, x_{v_2}^9 = 0, x_{v_3}^9 = 3.50, x_{v_4}^9 = 2.50, x_{v_5}^9 = 4.94, x_{v_6}^9 = 2.99, x_{v_7}^9 = 0$

$k = 10: x_{v_0}^{10} = 4.80, x_{v_1}^{10} = 2.50, x_{v_2}^{10} = 0, x_{v_3}^{10} = 3.50, x_{v_4}^{10} = 2.50, x_{v_5}^{10} = 4.94, x_{v_6}^{10} = 2.99, x_{v_7}^{10} = 0$

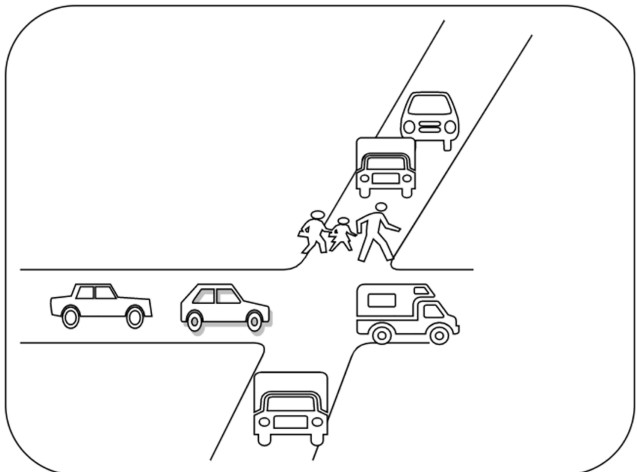

**Figure 3 A partitioned road environment, where an unmanned car runs autonomously without colliding with any of the pedestrians and other cars.**

According to this result, the expected cumulative reward for each state to reach $T$ can be obtained. For example, $ER(v_0 \models FT) = 4.80$, *i.e.*, in the state $v_0$, the expected cumulative reward of the robot (the system) to take the hazard item to region 4 is at least 4.80. Meanwhile, the environment always selects action $b$, which adopts the strategy to force the system with the minimum expected cumulative reward.

## Safety of autonomous driving

Consider the reward asynchronous probabilistic game $\mathscr{G}$ depicted in Fig. 3, it can be regarded as a simplified version of the unmanned car. The game models an unmanned car driving on an urban road and gives a route to navigate. There are three intersections $A$, $B$, $C$ in a road. When navigating the route, the car must autonomously and quickly react to dangers. Here, two dangers are considered: a pedestrian and a traffic jam. To avoid a traffic jam, the car changes the lane or honks. To avoid a pedestrian, the car brakes or honks. It is assumed that if the car goes through the first two intersections, it is safe to go through an intersection $C$.

The game graph corresponding to Fig. 3 is shown in Fig. 4. Consider an *RAPG* $\mathscr{G} = \langle \mathscr{V}, A_{ct}, V, P_e, P_s, L, R \rangle$, where $V_e = \{v_0, v_2, v_4, v_6, v_8\}$ is a set of the environment states, $V_s = \{v_1, v_3, v_5, v_7\}$ is a set of the system states, $A_{ct} = \{a, b, c, d, e, f\}$ is an action set, and $R = (R_{st}, R_{ac})$ is a synchronous reward structure. At the environment state, the environment can choose to appear a traffic jam and a pedestrian to prevent the car from reaching the intersection $C$ safely. These two actions are denoted as $a$ and $b$, respectively. Assume that the road is normal, and the environment has taken action $e$. At this time, the system can choose to brake, honk or change lane, these three actions are denoted as $c$, $d$, and $f$, respectively. Assume that the car is running normally, and the system takes action $g$. The process of the game is as follows. Starting from the state $v_0$, the car is running normally and at intersections $A$. If the environment takes action $a$, the

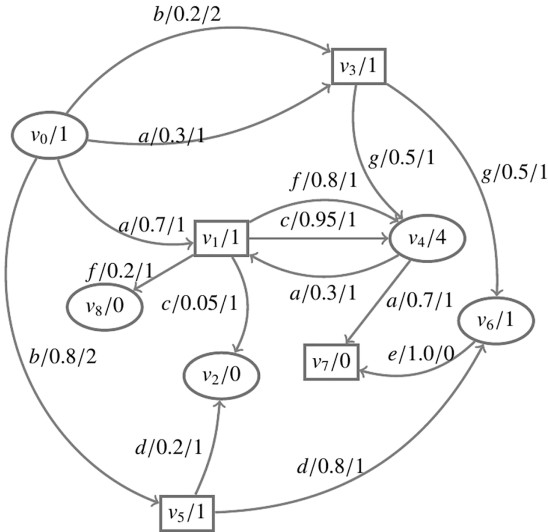

**Figure 4 The environment and the system are depicted as circles and squares, respectively.** The sum of the migration probabilities is equal to 1. $AP = \{succ, acc\}$ is the set of atomic propositions.

probability of a traffic jam is 0.7, and the probability of not causing a traffic jam is 0.3. If the environment chooses action $b$, the probability that a pedestrian appears is 0.8, and the probability that no pedestrian appears is 0.2. If the car (the system) takes action $c$, the probability of avoiding a traffic jam is 0.95, and the probability of not avoiding a traffic jam is 0.05. If the car (the system) takes action $f$, the probability of avoiding a traffic jam is 0.8, and the probability of not avoiding a traffic jam is 0.2. If the car (the system) takes action $d$, the probability of avoiding pedestrians is 0.8, and the probability of not avoiding pedestrians is 0.2. When the road is normal, the probability of normal driving of the car is 1. At this time, it is assumed that the system takes action $g$ and reaches the state determined by the environment with a probability of 0.5. Also, it is assumed that if the car safely reach intersection $A$ and $B$, it can safely reach intersection $C$. The following is the specific game diagram of the environment and the system, where $v_2, v_8$ is the accident state with the tag $acc$, and $v_7$ is the state when the intersection $C$ is reached safely. So, the set $T = \{v_7\}$ satisfies the winning condition $\psi$ and $G_\odot \neg T = \{v_2, v_8\}$ by Algorithm 1.

For the RAPG in Fig. 4, the asynchronous reward structure $R$ assigns $R(v_0) = R(v_1) = R(v_3) = R(v_4) = R(v_5) = R(v_6) = 1, R(v_2) = R(v_7) = R(v_8) = 0,$ $R(v_0, a) = 1, R(v_0, b) = 2, R(v_1, c) = R(v_1, f) = R(v_3, g) = R(v_4, a) = R(v_5, d) = 1,$ $R(v_6, e) = 0$. According to Algorithm 2, the expected cumulative reward to reach $T$ can be computed. Below, the partial values of $x_v^k$ for each state are presented:

$k = 0 : x_{v_0}^0 = 0, x_{v_1}^0 = 0, x_{v_2}^0 = 0, x_{v_3}^0 = 0, x_{v_4}^0 = 0, x_{v_5}^0 = 0, x_{v_6}^0 = 0, x_{v_7}^0 = 0, x_{v_8}^0 = 0$

$k = 1 : x_{v_0}^1 = 2, \ x_{v_1}^1 = 2, \ x_{v_2}^1 = 0, \ x_{v_3}^1 = 2, \ x_{v_4}^1 = 2, \ x_{v_5}^1 = 2, \ x_{v_6}^1 = 2, \ x_{v_7}^1 = 0, \ x_{v_8}^1 = 0$

$k = 2 : x_{v_0}^2 = 4, x_{v_1}^2 = 3.9, x_{v_2}^2 = 0, x_{v_3}^2 = 3.5, x_{v_4}^2 = 2.4, x_{v_5}^2 = 4, x_{v_6}^2 = 2.4, x_{v_7}^2 = 0, x_{v_8}^2 = 0$

$\cdots$

$k=13 : x_{v_0}^{13} = 6.10, x_{v_1}^{13} = 5.45, x_{v_2}^{13} = 0, x_{v_3}^{13} = 4.32, x_{v_4}^{13} = 3.64, x_{v_5}^{13} = 2.8, x_{v_6}^{13} = 1, x_{v_7}^{13} = 0, x_{v_8}^{13} = 0$

$k=14 : x_{v_0}^{14} = 6.10, x_{v_1}^{14} = 5.45, x_{v_2}^{14} = 0, x_{v_3}^{14} = 4.32, x_{v_4}^{14} = 3.64, x_{v_5}^{14} = 2.8, x_{v_6}^{14} = 1, x_{v_7}^{14} = 0, x_{v_8}^{14} = 0$

$k=15 : x_{v_0}^{15} = 6.10, x_{v_1}^{15} = 5.45, x_{v_2}^{15} = 0, x_{v_3}^{15} = 4.32, x_{v_4}^{15} = 3.64, x_{v_5}^{15} = 2.8, x_{v_6}^{15} = 1, x_{v_7}^{15} = 0, x_{v_8}^{15} = 0$

According to this result, the expected cumulative reward to reach the set $T$ can be obtained. For example, $ER(v_0 \models FT) = 6.10$, i.e., in the state $v_0$, the expected cumulative reward of the car (the system) to safely go through an intersection $C$ is at least 6.10. Meanwhile, in this state, a strategy that enables the car to reach the set $T$ can be derived, i.e., the environment takes action $b$.

In addition, during the experiment, it is found that the smaller the convergence threshold $\theta$, the larger the expected cumulative parameter $k$, and the more accurate the experimental results $x_v^k$.

## SUMMARY AND FUTURE WORK

This article studies reactive system synthesis problems and proposes a probabilistic model called reward asynchronous probabilistic games (RAPGs) for computing rewards in dynamic environments. Our model is motivated by players to choose actions through a reward mechanism, where the process generates rewards whose values depend on the state rewards and action rewards. The RAPGs are proposed with the LTL winning condition, which is a subclass of asynchronous probabilistic games. Meanwhile, the RAPGs can integrally describe the probabilistic behavior of the system and the environment. Besides, the synthesis algorithm is presented to compute the expected cumulative rewards. In addition, symbolic synthesis algorithms are provided for RAPGs to compute the maximum expected cumulative reward to satisfy the winning condition and synthesize the corresponding strategies. Our algorithm is formulated as a value iteration based on reinforcement learning.

Our proposed algorithm works as follows. First, the reachability properties are encoded by $\mu$-calculus formulas. According to the reachability properties, a set of states is obtained in which all plays starting from any state do not satisfy the winning condition. It is shown that the expected cumulative reward of any play that starts from any state of the set is 0. The method is clear, simple, and convenient. More importantly, it shrinks the state space of the algorithm that computes the expected cumulative reward. Then, an asynchronous reward mechanism is defined based on the winning condition of the RAPGs to incentive system to win. Based on this algorithmic reward construction procedure, a reinforcement learning algorithm is introduced to synthesize the optimal strategies that obtain the maximum expected cumulative reward for the system to win.

One interesting topic for further study is synthesizing strategies for mean-payoff rewards under the GR(1) winning condition. Another is to improve the scalability of the synthesis techniques to handle large and complex models. Meanwhile, we will extend the pattern of specifications and develop some tools to support automatic probabilistic synthesis.

### Funding

The authors received no funding for this work.

### Competing Interests

The authors declare that they have no competing interests.

### Author Contributions

- Wei Zhao conceived and designed the experiments, performed the experiments, analyzed the data, performed the computation work, prepared figures and/or tables, and approved the final draft.
- Zhiming Liu analyzed the data, authored or reviewed drafts of the article, and approved the final draft.

### Data Availability

The raw data and code is available at GitHub: https://github.com/wzhao618999/RL-for-RAPGs/tree/main/code.

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
