# Peer review of "A learning-based synthesis approach of reward asynchronous probabilistic games against the linear temporal logic winning condition"

_PeerJ Computer Science, doi:10.7717/peerj-cs.1094_

## Round 0.1 · original submission · Major Revisions

Based on 3 returned review reports, a major revision is needed for the current version. Basically, all the reviewers believe that the paper investigated an interesting and important topic. However, the background is not sufficiently introduced and the contribution has not been well-summarised to distinguish from the other existing results. The writing should be further improved as there are some citations missing in the paper, and the typos should be corrected in the revision.

Reviewer 1 ·

Basic reporting

This paper presents a learning-based synthesis approach of reward asynchronous probabilistic games against the linear temporal logic winning condition. The approach contains a completely automatic process that can generate strategies by analyzing the relations between the system and the environment, and provides an optimization to reduce the size of the output controller. Two running case studies demonstrate the effect of the synthesis approach. The proposed approaches appear to be sound. All underlying data have been provided; they are robust, statistically sound, and controlled. However, this paper also has some issues.
1. The main purpose of this method is to compute the maximum expected cumulative reward satisfying LTL specifications and generate corresponding strategies, but it is not compared with the other synthesis methods.
2. You mentioned LTL、\mu TL and CTL all together in the paper and mix-use them in your proposed approach. I consider this is rather unprofessional. Or do you want to propose a synthesis approach for a new logic that contains all features of LTL, \mu TL, and CTL?
3. The datasets used in this paper are relatively small, and is there a need for efficient validation of large data sets?

Experimental design

no comment

Validity of the findings

no comment

Additional comments

The author should check the full text for typos or grammatical errors and correct them. I suggest you have a colleague who is proficient in English and familiar with the subject matter review your manuscript, or contact a professional editing service.

Reviewer 2 ·

Basic reporting

This paper had on interesting topic. Although sufficient literature references and field background/context are provided, some related works are missing from the literature discussion. The proposed method is independent, with results are related to hypotheses. Figures are relevant, high quality, as well labelled clearly. The formal results also provide precise definitions of the theorems, as well as a detailed proofs. However, this paper also has some issues.
1.It is noted that your manuscript needs careful editing by someone with expertise in technical English editing paying particular attention to English spelling and sentence so that the goals and results of the study are clear to the reader.
2.Try to set the problem discussed on the manuscript in more clear.
3.Whether the experiment result is general characteristics?
4.If possible, explain how the data set of the experiment was selected?

Experimental design

Experiment results corroborate the proposed method. The research question is also clearly defined, pertinent, and meaningful.

Validity of the findings

The underlying experimental results have been provided in its entirety; they are robust, statistically sound. Conclusions are succinct, relevant to the original research question, and limited to supporting data.

Additional comments

The last sentence in the abstract replaces “two examples” by “two case studies”.

Annotated reviews are not available for download in order to protect the identity of reviewers who chose to remain anonymous.

Reviewer 3 ·

Basic reporting

This manuscript captures the maximum expected cumulative reward satisfying LTL specifications using a learning-based synthesis algorithm and proposes an asynchronous reward mechanism to motivate the system to satisfy specific requirements. This is an interesting paper with sound technical contributions. The manuscript is well organized though the presentation has room to be improved. The proposed method is demonstrated by two case studies. A more comprehensive evaluation of the method would be interesting to see. The manuscript must use clear, unambiguous, and professional English Language. The English language should be improved to ensure that an international audience can clearly understand your text. Some examples are as follows:
1. Page 6, line 84, authors should note the space between references and the text.
2. Page 7, line 96, “Markov decision processes (MDPs)”, I suggest that it be amended to “Markov Decision Processes (MDPs)”.
3. Page 12, line 292, authors should revise the typesetting of headlines and subheadings.
4. Page 15, line 346, authors should delete “ Example 2”.

Experimental design

1. How do ensure the correctness of the experimental results? Why is the strategy that satisfies the maximum expected cumulative reward extracted the optimal strategy?

Validity of the findings

no comment

Additional comments

1. What is the difference between the synthesis method proposed in the manuscript and the traditional synthesis algorithm? Why propose and study this question?
2. The contribution can be described in detail, where the motivation of the proposed method can be more specific.
3. The readability and presentation of the work should be improved. The font in the figure should be the same color and formal, e.g., Figure 2.

---

## Round 0.2 · accepted · Accept

The paper is well-revised based on the reviewers' comments. In the current submission, all the concerns have been addressed and the reviewers do not have further comments. Therefore, I believe that the paper is ready for publishing in PeerJ Computer Science, based on a solid contribution.

Reviewer 1 ·

Basic reporting

I am satisfied with the present manuscript.

Experimental design

no comment

Validity of the findings

no comment

Reviewer 2 ·

Basic reporting

This paper had an interesting topic. Although sufficient literature references and field background/context are provided, some related works are missing from the literature discussion. The proposed method is independent, with results are related to hypotheses. Figures are relevant, high quality, as well labelled clearly. The formal results also provide precise definitions of the theorems, as well as a detailed proofs.

Experimental design

Experiment results corroborate the proposed method. The research question is also clearly defined, pertinent, and meaningful.

Validity of the findings

Conclusions are succinct, relevant to the original research question, and limited to supporting data.

Reviewer 3 ·

Basic reporting

no comment

Experimental design

no comment

Validity of the findings

no comment